# Real-Time Short-Term Pedestrian Trajectory Prediction Based on Gait Biomechanics

**DOI:** 10.3390/s22155828

**Published:** 2022-08-04

**Authors:** Leticia González, Antonio M. López, Juan C. Álvarez, Diego Álvarez

**Affiliations:** Electrical Engineering Department, Campus of Gijon, University of Oviedo, 33204 Gijón, Spain

**Keywords:** motion trajectory prediction, kinematical models, gait biomechanics

## Abstract

The short-term prediction of a person’s trajectory during normal walking becomes necessary in many environments shared by humans and robots. Physics-based approaches based on Newton’s laws of motion seem best suited for short-term predictions, but the intrinsic properties of human walking conflict with the foundations of the basic kinematical models compromising their performance. In this paper, we propose a short-time prediction method based on gait biomechanics for real-time applications. This method relays on a single biomechanical variable, and it has a low computational burden, turning it into a feasible solution to implement in low-cost portable devices. We evaluate its performance from an experimental benchmark where several subjects walked steadily over straight and curved paths. With this approach, the results indicate a performance good enough to be applicable to a wide range of human–robot interaction applications.

## 1. Introduction

Human motion trajectory prediction (HMTP) is a critical technology in applications where people share their workspace with autonomous moving machines. That is needed, for example, in collaborative robotics for obstacle avoidance [1,2], in automatic driving assistance systems for safety assurance [3,4], in prostheses or exoskeletons for better performance [5], or in virtual reality to improve the sensation of immersion perceived by the user [6]. The strong interest in all these application fields explains the exponential growth of scientific communications devoted to this problem in the last few years [7].

A basic instance of the problem of HMTP can be described as how to estimate the future location of a specific mark in the body of a walking human within a given short temporal window (see Figure 1). With that prediction, the possibilities of enhancing intelligent human–robot collaborative environments increase. Robots can plan their motion to adjust their actions better for more efficient and safe collaboration with humans [8]. The prediction is based on information coming from the monitorisation of the human with environmental sensors, or sensors mounted on the robot, or wearables [9,10,11]. A viable real-time prediction of human trajectory must consider the sensors and signals that will be available and whether it will be possible to achieve the reactivity or fast response that the application demands.

As will be discussed below, there are different proposals in the literature for short-range anticipation of a walker’s position in real-time. All of them differ in both the amount of sensory information that needs to be provided and the type of information processing needed to solve the problem. In this paper, we propose a method with the following characteristics: (1) it is designed for close-proximity applications, with prediction windows around 1 s, (2) it relays on a single biomechanical variable that can be measured with inexpensive wearable sensors, and (3) it has a low computational load and is therefore feasible to implement in low-cost portable devices.

In the following section, we will present the relevant state-of-the-art. After describing the method details in Section 3, we evaluate its performance from an experimental benchmark where several subjects walked steadily over straight and curved paths, Section 4. We show that the main error source of the estimations has a specific biomechanical source: the left-to-right fluctuations of the position of the subject in the direction of the displacement induced by the alternation of the step. This effect is adequately compensated in our algorithm by introducing a sensor-based biomechanical compensation. The results in Section 5 indicate a performance good enough to be applicable to a wide range of human–robot interaction applications.

## 2. Related Work

Following the general taxonomy of HMTP tools proposed in [7], physics-based approaches based on Newton’s laws of motion seem best suited for real-time sensor-based short-term predictions because they operate within a reactive sense–predict scheme, avoiding intermediate processing time. For that reason, together with simplicity, we have ruled out methods that involved a higher level of cognition, such as learning based on data [12] or planning based on reasoning about motion intent [13].

Physics-based estimators, however, being useful to predict the motion of vehicles, fall short of capturing the quite complex dynamics of human walking. This has been addressed by resorting to a blending of a mixture of multiple dynamic models, but no performance improvement was found [14].

A similar strategy is to combine dynamic models with other learning or planning algorithms, even if, in principle, it goes against the speed of response of dynamic methods. A learning approach in [15] conveys promising results, but its performance on ambulatory motions such as walking has not been evaluated.

A third approach is to combine dynamic models with other information coming from the target agent himself. In [16], the short-term trajectory prediction is improved by tracking the user’s head in the context of virtual reality applications. Head orientation anticipates the trajectory, but it is also heavily influenced by other ambient-related impulses or distractions [17]. The authors suggest the use of eye trackers to overcome this problem, but this technology is too expensive to be used in more general intelligent environments. Our work is framed in this line but seeks to improve the prediction with the measurement of biomechanical signals that are easily implementable in practice.

## 3. Formulation of the Prediction Method

For the short-term prediction of the pedestrian, we will adopt a kinematic model and a sensor-based biomechanical compensation method as described below.

### 3.1. Kinematical Trajectory Prediction Models

Kinematical models are a good approach for the short-term prediction of the position of moving objects [18,19]. They are derived by applying the Newton laws from an initial position $\left(x,\;y\right)$ (see Figure 2), considering a set of initial conditions defined by the orientation of the displacement ϕ, the linear velocity v, the angular velocity w and the tangential acceleration $a_{t}$ as follows:(1)$$\dot{x}\left(t\right)=v\left(t\right)\cos\left(\phi \left(t\right)\right)$$
(2)$$\dot{y}\left(t\right)=v\left(t\right)sen\left(\phi \left(t\right)\right)$$
(3)$$\dot{\phi }\left(t\right)=w\left(t\right)$$
(4)$$\dot{v}\left(t\right)=a_{t}\left(t\right)$$

Depending on the assumptions taken about the translational and rotational velocities and the accelerations of the moving object, several particularisations can be formulated [18] (see Figure 3):(i)Constant Velocity (CV) model (w=a=0),(ii)Constant Acceleration (CA) model (w=0, a=constant≠0),(iii)Constant Turn Rate and Velocity (CTRV) model (w=constant≠0, a=0) and(iv)Constant Turn Rate and Acceleration (CTRA) model (w=constant≠0, a=constant≠0).

Defining the model state space vector as $\left[x,y,\phi ,v,a,w\right]\prime$, which includes the velocity vector $\left[v,\phi \right]\prime$, we can obtain the discrete version of the kinematical Equations (1)–(4). Assuming that in the time interval ∆t, the rotational velocity and the translational acceleration are constant (CTRA model) and results in:(5)$${\left[\begin{array}{c} x_{k+1}\\ y_{k+1}\\ \phi_{k+1}\\ v_{k+1}\\ a_{k+1}\\ w_{k+1} \end{array}\right]}_{CTRA}=\left[\begin{array}{c} x_{k}\\ y_{k}\\ \phi_{k}\\ v_{k}\\ a_{k}\\ w_{k} \end{array}\right]+\left[\begin{array}{c} \frac{a}{w_{k}^{2}\;}\cos\left(\phi_{k}+w_{k}∆t\right)-\frac{a}{w_{k}^{2}}\cos\left(\phi_{k}\right)+\frac{v_{k}+a∆t}{w_{k}}\sin\left(\phi_{k}+w_{k}∆t\right)-\frac{v_{k}}{w_{k}}\sin\left(\phi_{k}\right)\\ \frac{a}{w_{k}^{2}\;}\sin\left(\phi_{k}+w_{k}∆t\right)-\frac{a}{w_{k}^{2}}\sin\left(\phi_{k}\right)-\frac{v_{k}+a∆t}{w_{k}}\cos\left(\phi_{k}+w_{k}∆t\right)+\frac{v_{k}}{w_{k}}\cos\left(\phi_{k}\right)\\ w_{k}\cdot ∆t\\ a_{k}\cdot ∆t\;\\ 0\\ 0 \end{array}\right]$$

From this discrete CTRA model, the equations of the other three can be calculated by substituting their respective values of translational and rotational velocities and the accelerations:(6)$${\left[\begin{array}{c} x_{k+1}\\ y_{k+1}\\ \phi_{k+1}\\ v_{k+1}\\ w_{k+1} \end{array}\right]}_{CTRV}=\left[\begin{array}{c} x_{k}\\ y_{k}\\ \phi_{k}\\ v_{k}\\ w_{k} \end{array}\right]+\left[\begin{array}{c} \frac{v_{k}}{w_{k}}\sin\left(\phi_{k}+w_{k}∆t\right)-\frac{v_{k}}{w_{k}}\sin\left(\phi_{k}\right)\\ -\frac{v_{k}}{w_{k}}\cos\left(\phi_{k}+w_{k}∆t\right)+\frac{v_{k}}{w_{k}}\cos\left(\phi_{k}\right)\\ w_{k}\cdot ∆t\\ 0\\ 0 \end{array}\right]$$
(7)$${\left[\begin{array}{c} x_{k+1}\\ y_{k+1}\\ \phi_{k+1}\\ v_{k+1}\\ a_{k+1} \end{array}\right]}_{CA}=\left[\begin{array}{c} x_{k}\\ y_{k}\\ \phi_{k}\\ v_{k}\\ a_{k} \end{array}\right]+\left[\begin{array}{c} \left(v_{k}∆t+\frac{a}{2}∆t^{2}\right)\cos\left(\phi_{k}\right)\\ \left(v_{k}∆t+\frac{a}{2}∆t^{2}\right)\sin\left(\phi_{k}\right)\\ 0\\ a_{k}\cdot ∆t\;\\ 0 \end{array}\right]$$
(8)$${\left[\begin{array}{c} x_{k+1}\\ y_{k+1}\\ \phi_{k+1}\\ v_{k+1} \end{array}\right]}_{CV}=\left[\begin{array}{c} x_{k}\\ y_{k}\\ \phi_{k}\\ v_{k} \end{array}\right]+\left[\begin{array}{c} v_{k}\cos\left(\phi_{k}\right)∆t\\ v_{k}\sin\left(\phi_{k}\right)∆t\\ 0\\ 0 \end{array}\right]$$

At every sampling time, k, the model state vector $\left[x_{k},y_{k},\phi_{k},v_{k},a_{k},w_{k}\right]$, must be computed to apply it to the model. The actual orientation of the displacement $(\phi_{k}^{r}$) and the translational ($v_{k}^{r}$) are estimated from the position ($x_{k}^{r},y_{k}^{r}$):(9)$$\phi_{k}^{r}=\tan^{-1}\left(\left(y_{k}^{r}-y_{k-1}^{r}\right)/\left(x_{k}^{r}-x_{k-1}^{r}\right)\right)\;$$
(10)$$v_{k}^{r}=\mathrm{sqrt}\left({\left(x_{k}^{r}-x_{k-1}^{r}\right)}^{2}+{\left(y_{k}^{r}-x_{k-1}^{r}\right)}^{2}\right)/\Delta t$$

The rotational velocities ($w_{k}^{r}$) and translational acceleration ($a_{k}^{r}$) of the displacement are estimated from $\left(\phi_{k}^{r},v_{k}^{r}\right)$:(11)$$a_{k}^{r}=\left(v_{k}^{r}-v_{k-1}^{r}\right)/\Delta t$$
(12)$$w_{k}^{r}=\left(\phi_{k}^{r}-\phi_{k-1}^{r}\;\right)/\Delta t$$

In the following, we will refer to the sequence $\left(x_{k}^{r},\;y_{k}^{r},\;\phi_{k}^{r},\;v_{k}^{r},\;a_{k}^{r},\;w_{k}^{r}\right)$ as the raw trajectory since its variables are obtained recursively directly from the kinematic models, without any other consideration of the biomechanics of human walking. As discussed above, the predictive power of these equations alone can be expected to be unsatisfactory.

### 3.2. Offline Compensations

To correct the predictable limitations of the model discussed above, we will resort to studying the effects induced in it by gait biomechanics. It is known that in normal walking, the pelvis moves from side to side once per cycle, the trunk being over each leg to maintain balance. Similarly, the forward movement is not constant, and it produces variations in velocity according to the phase of the step. The resulting velocity and orientation signals corresponding to the pelvis have a sinusoidal shape [20,21].

To include that fact in the model, we propose offline filtering of the whole raw trajectory signals $\left(x_{k}^{r},y_{k}^{r}\right)$ to remove their waviness. We applied a polynomial regression, defining the offline estimated trajectories $\left(x_{k}^{b},y_{k}^{b}\right)$ for each of the experiments.

From the corrected position $\left(x_{k}^{b},y_{k}^{b}\right)$, we calculated the orientation of the displacement $(\phi_{k}^{b}$), the translational and rotational velocities ($v_{k}^{b}$ and $w_{k}^{b}$, respectively), and translational acceleration ($a_{k}^{b}$) of the displacement with previous Equations (9)–(12). By removing the undulations from the positions $\left(x_{k}^{b},y_{k}^{b}\right)$, the undulations will, in turn, be removed from the signals $\left(\phi_{k}^{b},v_{k}^{b},a_{k}^{b},w_{k}^{b}\right)$.

In the following, we will refer to the sequence $\left(x_{k}^{b},\;y_{k}^{b},\;\phi_{k}^{b},\;v_{k}^{b},\;a_{k}^{b},\;w_{k}^{b}\right)$ as the offline trajectory since its variables cannot be computed in real-time conditions. However, they could be taken as an instrumental way to test the utility of the proposed compensation.

### 3.3. Real-Time Sensor-Based Biomechanical Compensations

Assuming that the previous compensation will produce better estimations, it has the disadvantage of requiring filtering that induces time lags. We propose a real-time alternative to the offline compensation, which modifies the translational velocity and orientation signals $\left[v_{k}^{r},\;\phi_{k}^{r}\right]$ considering previous knowledge about gait biomechanics.

The global velocity of the centre-of-mass velocity of the body, $v_{k}^{r}$, is an oscillatory signal whose average value coincides approximately with the value it takes in the initial and final contacts of the feet, as described in gait studies [22]. Therefore, the displacement velocity can be better computed from signals taken from the initial contact of the ipsilateral foot (Figure 4(A)) to the subsequent final contact of the contralateral foot (Figure 4(B)), and then to the subsequent initial contact of the contralateral foot (Figure 4(C)).

This way, we can fix the raw displacement velocity, $v_{k}^{r}$ (Equation (10)), measured between initial and final contact foot events. We will refer to this sequence, ${\tilde{v}}_{k}$, as the real-time estimated velocity. Notice that the initial and final foot contacts can be detected from local maximums and minimums of the derivative of the raw velocity [23,24], so this correction does not require specific new sensors. The acceleration in these segments, ${\tilde{a}}_{k}$, can also be corrected and computed as the increment of velocity between consecutive events.

Regarding the orientation of the displacement $\phi_{k}$, it is known in gait biomechanics that the body trunk and the pelvis orientation evolve in counter-phase [21] (see Figure 5). Therefore, the raw position signal, $\phi_{k}^{r}$, can be corrected by measuring the pelvis orientation $\phi_{k}^{h}$. A proportional average can cancel the oscillations [25] and return the basic signal trend, ${\tilde{\phi }}_{k}$, closer to the actual forward orientation:(13)$${\tilde{\phi }}_{k}=\left(K\cdot \phi_{k}^{h}+\left(1-K\right)\;\phi_{k}^{r}\right)$$

The calibration gain *K* has to be experimentally calculated for each subject to minimise the least square error between the raw and the baseline trajectories. From the estimated orientation, $\phi_{k}$, we can compute the corresponding rotational velocity:(14)$${\tilde{w}}_{k}=\left({\tilde{\phi }}_{k}-{\tilde{\phi }}_{k-1}\;\right)/\Delta t$$

From the point of view of the real-time implementation, we have introduced the need for a sensor that measures hip orientation. As in the previous case of trunk velocity, it is possible to make this estimate with an IMU sensor placed on the hip [26].

In the following, we will refer to the sequence $\left(x_{k}^{r},\;y_{k}^{r},\;{\tilde{\phi }}_{k},\;{\tilde{v}}_{k},\;{\tilde{a}}_{k},\;{\tilde{w}}_{k}\right)$ as the estimated trajectory. This sequence defines the initial conditions for the application of the kinematical models in real time to forecast the position of the walking subject. From the discussion above, it is expected that this model will produce a better trajectory estimation than the raw trajectory and close to the offline trajectory.

In the following, we will test this proposed method to evaluate and compare the performance of the raw, offline estimated and real-time estimated trajectories.

## 4. Evaluation

### 4.1. Experimental Setup

For the evaluation of the models, an experimental benchmark was designed involving five adult subjects aged between 21 and 52. Two types of trajectories, *straight* and *curved*, were defined (see Figure 6). For the straight trajectories, participants walked in a straight line 5.4 m in length. Participants were instructed to perform a normal gait while maintaining a constant walking velocity. This trajectory was repeated five times per individual. For the curved trajectories, participants walk in circles around a central point, 1.5 m radius, again five repetitions. They were instructed to keep the turning radius as constant as possible with the help of a guide painted on the floor, increasing the speed as the experiment progressed.

An Optitrack system composed of 10 Flex3 cameras was used to monitor the experiments with a sampling frequency of 100 Hz. Calibration of the system was performed following the recommendations of the manufacturer, and the nominal residual errors achieved were 1.4 mm (mean). Following [27] (see Figure 7), we used the bidimensional raw position $\left(x_{k},\;y_{k}\right)$ of the centroid of the rigid body formed from five markers placed around the waist as the subject position. Pelvis orientation, $\phi_{k}^{h}$, was estimated from the actual orientation of the rigid body from the defined reference system. A 3rd-order low-pass Butterworth filter with a cut-off frequency of 6 Hz was applied during the acquisition to remove frequential components above those pertinent for the human gait.

### 4.2. Model Application and Error Analysis

Data from the curved experiment was divided into five trajectory segments corresponding to each of the individual turns. This way, we defined for each subject five straight trajectory segments and five curved trajectory segments (each corresponding to a whole turn). To form the offline estimated trajectories, we use linear regression for straight segments and for the curved segments, a grade 8 polynomial regression.

The four prediction models (CV, CA, CTRV, CTRA) were then sequentially applied to the sequence $\left(x_{k}^{r},\;y_{k}^{r},\;{\tilde{\phi }}_{k},\;{\tilde{v}}_{k},\;{\tilde{a}}_{k},\;{\tilde{w}}_{k}\right)$ starting at every sample from k=1 to k=N−100, being N the length of the current trajectory data. Each sample is predicted to have a horizon of 1 s (100 samples) (see Figure 8). For this purpose, the models are applied recursively by introducing as input the output of the previous iteration until the 1-s horizon is raised.

Similarly, these models are applied directly to the signals without removing the wavelets $\left(x_{k}^{r},\;y_{k}^{r},\;\phi_{k}^{r},\;v_{k}^{r},\;a_{k}^{r},\;w_{k}^{r}\right)$ and on the post-processed signal with offline compensation $\left(x_{k}^{b},\;y_{k}^{b},\;\phi_{k}^{b},\;v_{k}^{b},\;a_{k}^{b},\;w_{k}^{b}\right)$.

For every application of a model starting at sample k, the prediction error sequence $e_{k}$ was defined as the sequence of Euclidean distances between the predicted $\left({\hat{x}}_{k\ldots k+99},\;{\hat{y}}_{k\ldots k+99}\right)$ and actual position values $\left(x_{k\ldots k+99}^{r},y_{k\ldots k+99}^{r}\right)$ of the raw trajectory.

## 5. Results

Figure 9 shows the Root Mean Square of the error sequences (predictions from one sample to one hundred samples, i.e., 0.01 s to 1 s) for each model (CV, CA, CTRV, CTRA) for the raw trajectories (top), offline estimated trajectories (centre), and real-time estimated trajectories (bottom).

We have found that prediction errors present an exponential growth with time, characteristic of the usual drift present in estimations based on integral procedures, with different growth rates for the different models. RMS prediction error at 1-s length is the biggest for predictions from the raw trajectories, with RMS error of predictions from offline estimated trajectories the lowest in general.

Table 1 contains the RMS error at a one-second prediction length from each trajectory (raw, offline estimated, and real-time estimated) and each prediction model. In general, as could be expected, curved models (CTRV, CTRA) perform better for curved trajectories, and straight models (CV, CA) perform better for straight trajectories. We found an exception in predictions from offline estimated trajectories, where there was no difference between the four prediction models for the straight trajectories. The translational acceleration and rotational velocity estimated for these trajectories were almost zero, as planned with the design of the experiment, which implies that all models adopted the behaviour of the CV model. Results also show that accelerated models perform in general worse than non-accelerated models, especially in the raw trajectories. In aggregated terms, the CTRV performs the best in all situations. Considering the type of path (straight, curved), the CTRV still performs the best except for prediction for straight real-time estimated trajectories, where straight models (CV, CA) perform better than the CTRV model.

To further analyse the performance of the prediction models, a multiway analysis of the variance (ANOVA) was carried out using MATLAB^®^ R2020b from MathWorks to check if the mean prediction at a horizon of prediction of 1 s was affected by the intervening factors: the subject performing the test, the type of path (straight or curved), and the model applied for the prediction (CV, CA, CTRV, CTRA). The subject factor was treated as a random effect, while the type of path and model factors were treated as a fixed effect.

The results showed that mean prediction errors were not affected by the subject performing the test (*p*-value > 0.01) for the raw, offline estimated, and real-time estimated trajectories. On the contrary, the type of path and the prediction model significantly affected the mean prediction error for the raw, offline estimated, and real-time estimated trajectories (*p*-value < 0.01). A multiple comparison test was then carried out to analyse significant differences in the mean prediction error for pairs of type of trajectory/model.

For the offline estimated trajectories, we found four groups with mean prediction errors that were not significantly different among them and were significantly different from errors from the other three groups. These groups, expressed in ascending order or the averaged prediction error, were:Group 1: CV and CA in straight segments.Group 2: CTRV and CTRA in straight segments.Group 2: CTRV and CTRA in curved segments.Group 3: CV and CA in curved segments.

For the real-time estimated trajectories, we identified two different groups:Group 1: CV in straight segments; CTRV in straight and curved segments.Group 2: CA in straight segments; CTRA in straight and curved segments.

Regarding the raw trajectories, the test showed that the mean prediction error was significantly different for all type of trajectory/model pairs.

## 6. Discussion

### 6.1. Experiments Design

Experiments were designed to evaluate the performance of the proposed model during normal walking and, therefore, they comprised walks in straight and curved trajectories. We designed experiments for the extreme cases, straight trajectories at a constant velocity and circular trajectories with increasing velocity of displacement. The prediction results of the proposed model are compared with those obtained by applying the kinematic models directly to the raw data and to the post-processed data to be able to quantify the improvement achieved.

### 6.2. Impact of Velocity and Orientation on Predictions

Kinematical models based on the well-known Newton Laws have been used in the state-of-the-art as a general approach to forecasting the position of general moving objects. For instance, in [28], the authors perform a comparative study of several dynamical models (CV, CA, CTRV) and a multi-model algorithm combining such basic models. Particularly in terms of single motion models, they observe benefits for the CV model in some scenes and CTRV in others, making complex models inefficient. Similarly, we have observed that these models have usually been applied as a complement to other advanced techniques [15]. For our proposal prediction method, we chose to use the four models describing four types of trajectories where the occurrence of translational accelerations and angular velocities are combined.

The results of applying these models to our raw data confirm how the left-to-right fluctuations of the position of the subject in the direction of the displacement induced by the alternation of the step affect the performance of the basic kinematical models. Prediction on the raw data has the worst performance. On these, we can distinguish two effects. Firstly, the RMS of the accelerated models is significantly higher than that of the non-accelerated models for both curves and straight lines, in contrast to the offline prediction. This is the result of the influence of using the sinusoidal velocity in the kinematic models. A similar effect can also be found for straight versus curved models on straight trajectories. While in the offline prediction, all models have the same mean error, with the raw data, the CTRV model outperforms the RMS of the CV and the CTRA of the CA. This is due to the oscillation of the orientation, which is picked up by the curvilinear models.

### 6.3. Impact of Basic Kinematic Models on Predictions

The best model to apply in a general situation is the CTRV model. It shows the best prediction errors in aggregated terms. We have also found that the use of accelerated models supposes a concern. On the one side, we have found that for offline curved trajectories, curved models, accelerated and non-accelerated (CTRV, CTRA) performed equally. This finding was unexpected, as subjects were told to accelerate during the development of the curved experiments to analyse the performance of accelerated models (CA, CTRA) compared to non-accelerated models (CV, CTRV). However, we found that the acceleration estimated from the curved baseline trajectories ($a_{k}^{b}$) was on average 0.04 ± 0.09 m/s^2^, coherent with reference data for normal walking reported in the literature [9]. This value was possibly too small to make a difference in performance between the two groups of models, as confirmed by the multiple comparison test. Therefore, this finding supports the idea that accelerated models do not provide a real improved performance over non-accelerated models in the context of predicting position during human walking. Moreover, the use of the acceleration in the models may suppose an issue, as the estimated prediction seems to present a higher error to this parameter, even with the velocity compensation.

In general terms, for the reasons stated above, the CTRV performs the best for unconstrained trajectories. We provide a statistical analysis that reveals the model most efficient for each type of path. In case the type of path is known in advance, the taxonomy of models may help to choose the most appropriate for a given application framework.

### 6.4. Performance of Real-Time Sensor-Based Compensation Method

The increased performance of predictions from offline filtered trajectories compared to predictions from raw trajectories confirms the necessity of a technique to remove the fluctuations from the raw signals before applying the prediction kinematical models. For this work, we have made batch filtering considering the whole raw signals to define the offline filtered trajectories. However, in a real application scenario of prediction of the subject position, such undulations should be removed in real-time, considering only the signal sampled to the actual moment. Future positions of the subject would not be available, and processing like the one performed to define the baseline trajectories would not be possible.

Classical frequential filters would be an option to remove the gait-induced undulations found in the sampled signals. In certain situations, these filters could perform properly. For instance, if the trajectory is essentially straight or the velocity is essentially constant, a low pass filter of a few seconds in length could be adequate. However, they may present some drawbacks, as the low step frequencies of human gait (between 0.74 and 1.3 two-step/s [29]) would require slow filters that would introduce a delay too high for a short-time prediction (a four-pole Butterworth filter with a cutting frequency of 1 Hz is expected to add a delay about 0.5 s [30]). So, for changing orientations and eventually changing velocities, these filters could not be very efficient. Recent results [25] confirm that this approach may lead to erroneous results and that a different approach is needed for a general solution to the problem.

In this paper, we propose to address the elimination of the undulations in the orientation and the velocity from the consideration of the biomechanical characteristics of human gait. We aimed to provide a general approach applicable to unconstrained trajectories mixing straight and curved paths and eventually different linear and rotational velocities and accelerations. With this approach, we show how prediction errors are considerably reduced from predictions from raw trajectories, making them comparable to the prediction from baseline estimations. Anyway, there is still room for improvement. Regarding the use of the acceleration in the models, it is true that the biomechanical inspired corrections made over the acceleration reduce the neglecting effect they present of the prediction from the raw trajectories. However, as acceleration makes a big impact on error and acceleration in human walking is low, the use of accelerated models may not be very profitable.

Comparing the results with those of other works, we find results in the same range of values. An example is the case of [15], where the proposed prediction method achieves an average error between 100 and 200 mm for predictions at a horizon of 0.5 s. The result obtained by individually applying a CV model (called as Velocity-Based Position Projection Method) gives a mean error of 250 mm. In our case, the RMS for the four kinematic models at a prediction horizon of 0.5 s is between 50 and 100 mm, as opposed to an RMS of ≈120 mm by the CV model applied to the raw data.

## 7. Conclusions

In a human–robot collaborative space, the short-term prediction of a person’s walking position becomes necessary from a real and perceived safety point of view. In this paper, we propose a human trajectory prediction method for real-time application based on the biomechanical characteristics of human gait. This approach can be applied with inexpensive wearable sensors, and it has a low computational load. Experiments were designed to evaluate the performance of the proposed method during normal walking, comprising straight and curved paths. The results confirm how the left-to-right fluctuations of the position of the subject in the direction of the displacement induced by the alternation of the step affect the performance of the basic kinematical models to predict by themselves. We have found that if such undulations are removed, prediction performance would be improved and propose a prediction method that compensates the body swing by correcting velocity and orientation based on the initial and final contacts of the feet and the pelvic motion. The results show that, as acceleration makes a big impact on error and acceleration in human walking is low, the use of accelerated models may not be very profitable. Despite this, the performance of the proposed prediction method improves the use of basic kinematical models and produces results compatible with real-time applications.

## Figures and Tables

**Figure 1 sensors-22-05828-f001:**
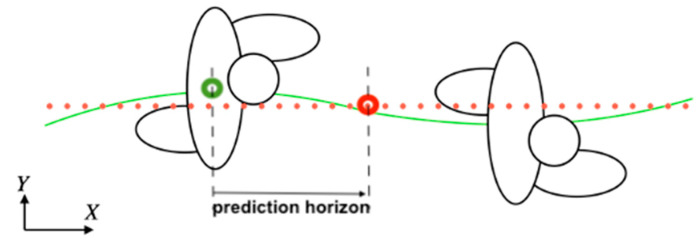
An instance of the problem of human motion trajectory prediction: the location of a specific landmark in the walking human body (green dot) is projected to its future location (red dot) within a given prediction horizon. The overall person’s mark movement (solid green line) results in an estimated walk trajectory (red dots line).

**Figure 2 sensors-22-05828-f002:**
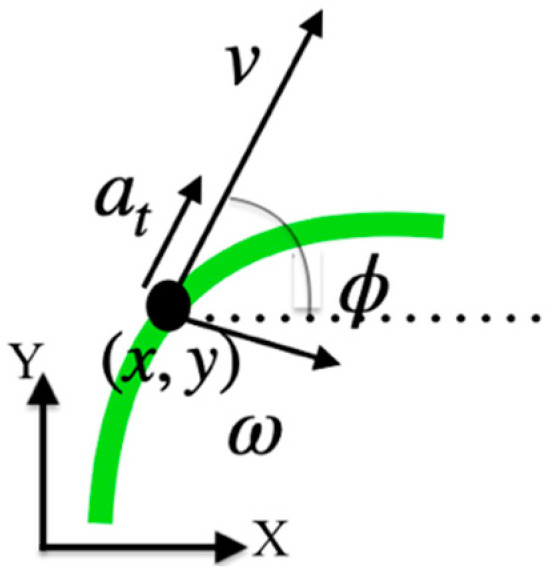
Geometry of motion for the 2-dimensional kinematical model of a moving object.

**Figure 3 sensors-22-05828-f003:**
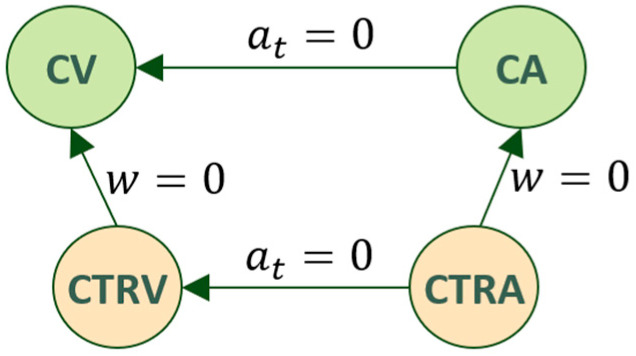
Relationship between common movement patterns defined from simplifying assumptions over the general model: CV (straight displacement at Constant Velocity), CA (straight displacement at Constant Acceleration), CTRV (curved displacement, at constant displacement and rotational velocities), and CTRA (curved displacement, at constant rotational velocity and constant accelerated displacement).

**Figure 4 sensors-22-05828-f004:**
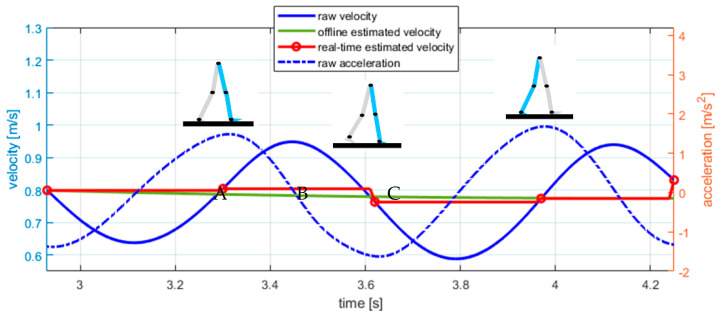
Forward real-time estimated velocity (red line) and offline estimated velocity (green line). The estimation of the forward velocity is addressed by holding the actual value of the raw velocity (continuous blue line) at the initial contact of the ipsilateral foot until the final contact of the contralateral foot and vice–versa. Initial and final contacts are detected from local maxima and minima of the derivative of the velocity (dashed blue line).

**Figure 5 sensors-22-05828-f005:**
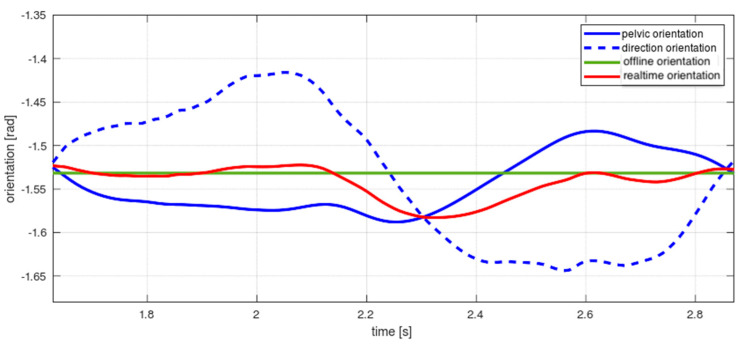
Orientation estimation (red line) and offline estimated orientation (green line). The real-time estimation is based on a proportional combination of the counter–phase signals pelvic orientation (continuous blue line) and the orientation estimated from the sampled raw spatial position of the subject (dashed blue line).

**Figure 6 sensors-22-05828-f006:**
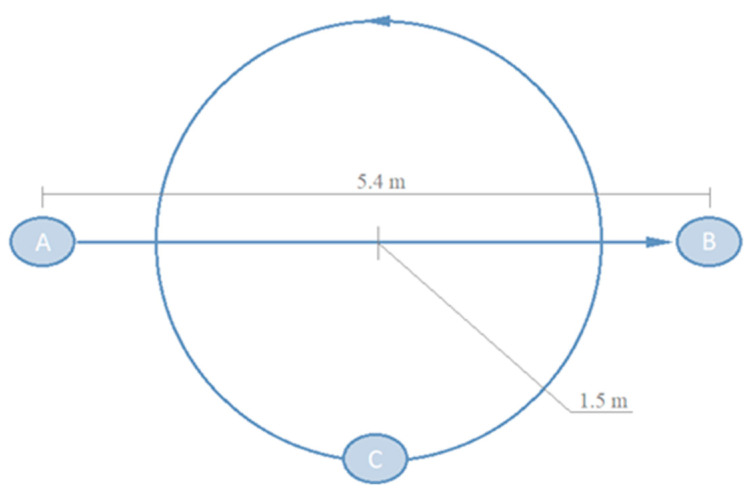
Experiments involved displacements over straight (5.4 m length) and circular trajectories (1.5 m radius). Straight trajectories ran from point A to B. For curved trajectories, participants describe a circle starting and ending at point C.

**Figure 7 sensors-22-05828-f007:**
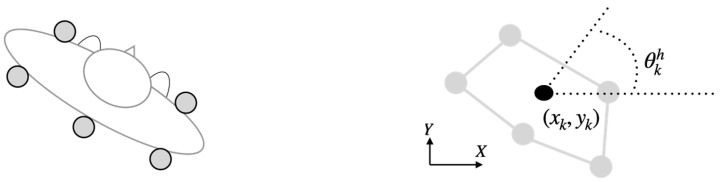
Five reflective markers were placed around the waist of the subject (**left**) for the estimation of the 2-dimensional spatial position of the body $\left(x_{k},\;y_{k}\right)$ and the orientation $\left(\theta_{k}^{h}\right)$ of the pelvis (**right**).

**Figure 8 sensors-22-05828-f008:**
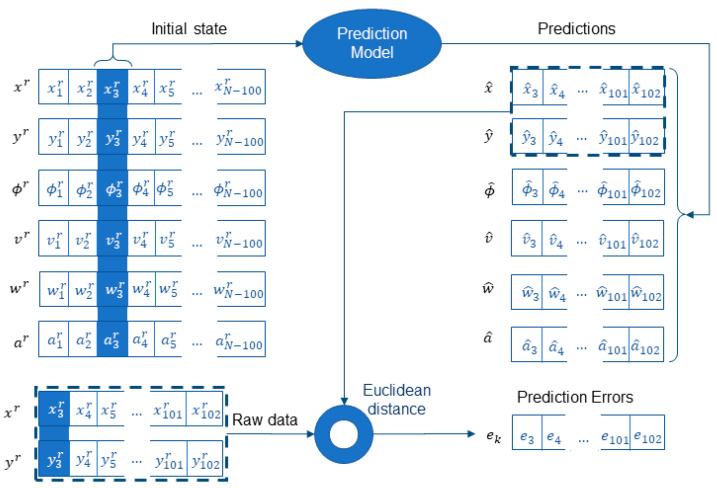
The figure describes the application of a prediction model starting at the third sample (k=3) of a raw trajectory $\left(x_{k}^{r},\;y_{k}^{r},\;\phi_{k}^{r},\;v_{k}^{r},\;a_{k}^{r},\;w_{k}^{r}\right)$. Predictions are sequentially extended for one second (100 samples, k=3…102) using the initial state vector highlighted in the blue column (**left**). The prediction error sequence is then defined from the Euclidean distance of predicted and actual positions of the subject (k=3…102).

**Figure 9 sensors-22-05828-f009:**
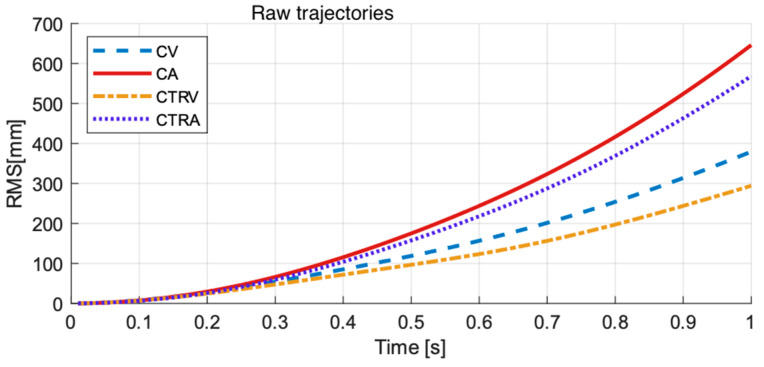
RMS prediction errors up to a prediction horizon of 1 s from the raw (**top**), offline estimated (**centre**), and real-time estimated (**bottom**) trajectories.

**Table 1 sensors-22-05828-t001:** RMS error [mm] of the prediction on the theoretical offline estimated, raw and real-time estimated trajectories at a one-second prediction horizon. Results are shown for each prediction model and each type of trajectory (straight/curved). Aggregated values show the RMS prediction error considering straight and curved trajectories together.

		CV	CA	CTRV	CTRA
Offline estimated	Aggregated	360.83	363.80	78.15	82.83
	Curved	445.13	448.82	92.1	98.17
	Straights	40.12	40.12	40.12	40.12
Raw	Aggregated	379.52	646.16	294.27	568.33
	Curved	454.5	718.52	305.17	579.13
	Straights	160.28	480.28	272.44	547.31
Real-time estimated	Aggregated	357.5	369.04	113.33	130.59
	Curved	434.66	447.78	106.2	127.94
	Straights	110.2	120.29	125.73	135.44

## Data Availability

The data presented in this study are available on request from the corresponding authors.

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
