# Peer review of "Real-Time Short-Term Pedestrian Trajectory Prediction Based on Gait Biomechanics"

_sensors, 2022, doi:10.3390/s22155828_

Round 1
Reviewer 1 Report
The authors of this paper presented a prediction model for the short-term of the trajectory of a person during normal walking. They evaluated for that purpose the performance in real-time of the fundamental kinematical trajectory prediction models from an experimental benchmark where subjects walked steadily over straight and curved trajectories. The results show that the intrinsic properties of human walking conflicts with the foundations of the kinematical models compromising their performance. They showed how the gait left-to-right fluctuations of the position of the subject in the direction of the displacement induced by the alternation of the step affect the estimation of the actual conditions of the displacement limiting the capacity of the canonical models to forecast the position of the walking subject.
It is very nice paper that can be considered for publication with some major and minor changes, as follows:
-First, the novelty must be clarified. Describe well your new contribution in both abstract and conclusion.
- How did you consider the parameter settings of the applied methods? Did you try other settings?
- Improve the quality of some figures, the text and notations can not be read in some figures.
- Clarify the data preprocessing and training and testing scenarios.
- Add more details about the applied methods.
- Refresh your study with some recent studies, such as Sensor-Based Human Activity Recognition with Spatio-Temporal Deep Learning; Multi-ResAtt: Multilevel Residual Network with Attention for Human Activity Recognition Using Wearable Sensors; A Context-Based Multisensor Sensor Data Fusion Algorithm for the Internet of Things; Multi-sensor human activity recognition using CNN and GRU;
-The limitations must be discussed.
- recheck all equations and notations.
Author Response
Motivated by your comments and indications, we thought that the presentation of the paper was flawed on our part. The focus and intent of the paper were not clear, and we believe that this has made the work of the reviewers difficult.
We regret that we did not get this right and have therefore decided to proceed with a complete rewrite of important parts of the paper, especially those referring to the definition of the paper, the state of the art, related works and references, and the motivation.
Concerning the comments, we make the following changes:
- First, the novelty must be clarified. Describe well your new contribution in both abstract and conclusion.
We rewrote the abstract, the introduction and the conclusions in such a way as to highlight the contribution of the article. This needed to be refocused to clarify that the contribution of the article is a new method of predicting a person's walking trajectory based on the conditions of human walking itself. To evaluate this method, it is compared with the individual application of 4 basic kinematic methods, as well as with a method with offline filtering.
- How did you consider the parameter settings of the applied methods? Did you try other settings?
At line 195, we explain the way to adjust the calibration gain K, which it is dependent on each subject. We studied the use of a general K (reference [25]), but it did not fit all people correctly. We therefore decided to use an independent K adapted to each of the subjects.
- Improve the quality of some figures, the text and notations can not be read in some figures.
We have redone most of the graphs, both to adjust them to the new nomenclature we use and to improve their visualisation.
- Clarify the data preprocessing and training and testing scenarios. - Add more details about the applied methods
We have also reordered and rewritten sections 3 and 4 on methods and method evaluation. We hope that we have been able to improve their understanding.
- Refresh your study with some recent studies, such as Sensor-Based Human Activity Recognition with Spatio-Temporal Deep Learning; Multi-ResAtt: Multilevel Residual Network with Attention for Human Activity Recognition Using Wearable Sensors; A Context-Based Multisensor Sensor Data Fusion Algorithm for the Internet of Things; Multi-sensor human activity recognition using CNN and GRU;
We add after the introduction a section of related work to introduce the reader to the currently existing alternatives in human trajectory prediction, trying to use more recent references. This section is made from the point of view of a prediction for a real-time application that requires high reactivity and simplicity in data collection.
- The limitations must be discussed.
In the discussion, also rewritten and now divided into sub-sections, we discuss the limitations of the proposed method, as well as the problems encountered when applying the basic kinematic models on their own, and the conflict of trying to correct their application with very aggressive filtering techniques that lead to high delays.
- recheck all equations and notations
We adapted the equations as necessary to the new nomenclature and explanations. We also eliminated some errors in the editing of the equations that could make them difficult to read in Word documents.
We hope that this major revision will serve to better explain where we see the contribution of our work and put into context the experimental results presented.
We thank the reviewer for his sincerity, which has motivated us for this major revision, and which we believe substantially improves the final result.
Reviewer 2 Report
Dear authors,
Thanks for your contribution to Sensors.
Before further process of this manuscript, please check if it matches the scope of the journal.
With major revisions of the manuscript, it might be reconsidered.
The opinions are set out below:
STRUCTURE
Please prepare the manuscript following the instructions for authors.
ENGLISH
The manuscript has several typos. Authors need to proofread the paper to eliminate all of them. Some sentences are too long. Generally, it is preferable to write short sentences with one idea in each sentence.
REFERENCES
The literature review is incomplete. Several relevant references are missing. The reference list should include the full title, as recommended by the style guide.
INTRODUCTION
Authors should include additional references in the introduction that support the claims. Authors should better explain the background to this research, including why the research issue is important. Contributions should be enhanced. It should be made clear what is novel and how it addresses the limitations of prior work.
RELATED WORK
The related work section is not well organized. Writers should try to categorize articles and present them logically. Authors should add a table comparing the main features of previous work in order to highlight their differences and limitations. Alternatively, authors may consider adding a row to the table to describe the proposed solution.
PROBLEM DEFINITION
Authors should provide a clear and detailed definition of the issue. Authors should include an example to illustrate how the problem is defined.
METHOD
A novel solution is presented, but it is important to better explain the design decisions (e.g. why the solution is designed that way). There is a need for discussion of the complexity of the proposed solution.
EXPERIMENT
The experiments should be updated to incorporate some comparisons to newer studies. Please provide a description of why only 5 participants participated in the study.
Sincerely yours,
Author Response
Thank you for the comments and the indications.
Motivated by the same, we thought that the presentation of the paper was flawed on our part. The focus and intent of the paper were not clear, and we believe that this has made the work of the reviewers difficult.
We regret that we did not get this right and have therefore decided to proceed with a complete rewrite of important parts of the paper, especially those referring to the definition of the paper, the state of the art, related works and references, and the motivation. All these sections called for improvements, which we believe we have made.
We also hope that this major revision will serve to better explain where we see the contribution of our work and put into context the experimental results presented.
We thank the reviewer for his sincerity, which has motivated us for this major revision, and which we believe substantially improves the final result.
Round 2
Reviewer 1 Report
The authors addressed all comments. This paper can be accepted.
Reviewer 2 Report
Accept in present form